# Characterization of the Basal and mTOR-Dependent Acute Pulmonary and Systemic Immune Response in a Murine Model of Combined Burn and Inhalation Injury

**DOI:** 10.3390/ijms23158779

**Published:** 2022-08-07

**Authors:** Hannah R. Hall, Cressida Mahung, Julia L. M. Dunn, Laurel M. Kartchner, Roland F. Seim, Bruce A. Cairns, Shannon M. Wallet, Robert Maile

**Affiliations:** 1Department of Pathology and Laboratory Medicine, University of North Carolina at Chapel Hill, Chapel Hill, NC 27599, USA; 2North Carolina Jaycee Burn Center, Department of Surgery, University of North Carolina at Chapel Hill, Chapel Hill, NC 27599, USA; 3Department of Microbiology and Immunology, University of North Carolina at Chapel Hill, Chapel Hill, NC 27599, USA; 4Curriculum in Toxicology and Environmental Medicine, University of North Carolina at Chapel Hill, Chapel Hill, NC 27599, USA; 5Division of Oral and Craniofacial Health Sciences, University of North Carolina Adams School of Dentistry, Chapel Hill, NC 27599, USA

**Keywords:** burn injury, inhalation injury, mTOR

## Abstract

Severe burn injury leads to a cascade of local and systemic immune responses that trigger an extreme state of immune dysfunction, leaving the patient highly susceptible to acute and chronic infection. When combined with inhalation injury, burn patients have higher mortality and a greater chance of developing secondary respiratory complications including infection. No animal model of combined burn and inhalation injury (B+I) exists that accurately mirrors the human clinical picture, nor are there any effective immunotherapies or predictive models of the risk of immune dysfunction. Our earlier work showed that the mechanistic/mammalian target of rapamycin (mTOR) pathway is activated early after burn injury, and its chemical blockade at injury reduced subsequent chronic bacterial susceptibility. It is unclear if mTOR plays a role in the exacerbated immune dysfunction seen after B+I injury. We aimed to: (1) characterize a novel murine model of B+I injury, and (2) investigate the role of mTOR in the immune response after B+I injury. Pulmonary and systemic immune responses to B+I were characterized in the absence or presence of mTOR inhibition at the time of injury. Data describe a murine model of B+I with inhalation-specific immune phenotypes and implicate mTOR in the acute immune dysfunction observed.

## 1. Introduction

Burn injuries lead to significant numbers of hospitalizations and deaths every year in the United States, costing over USD 5.5 billion between hospitalization and treatment expenses and loss of income [1]. Burn patients experience one of the most severe forms of injury [2], often necessitating longer hospital stays than any other trauma [3,4]. There are multiple influences on morbidity and mortality in burn patients, with inhalation injury among the most significant [5,6]. Combined burn and inhalation injury (B+I) occurs in 5–30% of all burn patients and is characterized by epithelial denudation, elevated leukocyte (neutrophil and macrophage) activity in the lung, and enhanced local and systemic inflammation which, when combined, lead to increased morbidity and mortality of burn patients attributed to increased lung damage and bacterial infections [7]. Although the treatment of burn wounds has significantly advanced over recent years [8], smoke inhalation injury has proven difficult to treat and remains associated with increased mortality and morbidity [9,10].

Inhalation injury and burn injury have independently been shown to mediate significant immune dysregulation [11,12,13,14,15,16,17]. Studies of patient populations indicate that each injury independently causes immune dysregulation, and that their combined effects result in damage in excess of that expected due to individual contributions from each injury [18]. For instance, we and others have observed that either burn injury or inhalation injury alone can lead to increased neutrophil presence in the lung tissue compartment [19,20,21,22,23]. Neutrophils within the bronchial spaces have been associated with immune dysfunction (such as hyperexpression of MCP-1 and IL-6 and increased reactive oxygen and nitrogen species) and increased susceptibility to bacterial infection [24,25,26]. We have also shown that in both mice and rats, smoke inhalation injury alone is similarly associated with increased cytokine and chemokine production, and oxidative stress in the presence of increased neutrophils, resulting in subsequent damage to the pulmonary compartment [22,27,28,29]. Burn or inhalation injury alone results in an accumulation of immune cells into the lung cavity with increased antimicrobial activity late after injury [30]. Although combined B+I injury is known to cause significant complications and lead to a worsened outcome, few models have been generated that examine the immune-mediated effects of combined B+I injury [19,31]. In addition, although it is clear that burn injury induces a “genomic storm” after injury [32,33], and gene groups have been described that were impacted by injury, no specific mechanisms have been defined, nor therapies exist, for burn injury that specifically address the dysfunctional immune response [34]. 

An attractive mechanistic target to investigate conditions of immune dysfunction is the mechanistic/mammalian target of rapamycin (mTOR), which has been directly linked with immune dysfunction and pulmonary damage after injury [35,36,37]. Burn injury and inhalation injury both generate damage and the release of numerous inflammatory stimuli (damage-associated molecular patterns (DAMPs)) that resident innate immune cells (e.g., neutrophil and macrophages) and noncanonical immune responder cells (e.g., pulmonary epithelial cells) bind to through pattern recognition receptors (PRR), including toll-like receptors (TLR). TLR signaling results in the activation of the mTOR pathway which drives immune processes, including those associated with proinflammatory cytokine expression that results in recruitment and activation of immune cells promoting exacerbated tissue damage and the clinically described systemic inflammatory responses syndrome (SIRS) [38]. As part of a negative feedback mechanism, mTOR activation also leads to upregulation of the negative regulator of inflammation, PPARγ, which in turn represses immune responses [39,40]. We have previously demonstrated that mTOR activation is observed in pulmonary neutrophils and macrophages three days after cutaneous burn injury and that inhibition of mTOR at the time of injury with rapamycin, a well-characterized mTOR inhibitor, prevents acute pulmonary neutrophil recruitment, acute pulmonary and systemic innate cell activation, and induces chronic bacterial susceptibility compared to vehicle-treated burn-injured controls [41].

In this study, we describe a novel preclinical murine model of combined B+I injury that mirrors the increased clinical severity of combined injuries, and that we hypothesize could be used to dissect the systemic and pulmonary genes that are responsible for the immune dysfunction observed early after injury. Indeed, we demonstrate that combined B+I injury leads to immune dysfunction and that the immune consequences of this combined injury are more pronounced than either injury alone, and we identify specific immune gene signaling changes based on injury modality. We then use this model to define the effect of the systemic mTOR blockade on systemic and pulmonary immune outcomes early and late after combined injury.

## 2. Results

### 2.1. Combined Burn and Smoke Inhalation (B+I) Injury Leads to Higher Mortality than Burn Alone, with Elevated Levels of Protein and Cellular Infiltrate within the Airway Space and Lung Tissue

We combined our mouse models of cutaneous burn injury [42] with our model of woodsmoke inhalation injury [43] and demonstrated an increased insult of injury when compared to burn injury alone or sham injury. The addition of smoke inhalation to burn injury results in a ~50% survival rate after 14 days, compared to ~90% survival in burn and sham mice (Figure 1A). To evaluate the extent of lung damage, total proteins were evaluated within the bronchoalveolar lavage fluid (BALF), which revealed that protein concentration within the lung cavity was significantly increased in B+I mice compared to burn mice at 24 h and at 14 days post-injury (Figure 1B), likely due to increases in lung damage. In addition, upon evaluation of the cellular infiltration of the lung tissue, we observed that the total cell numbers within the lung tissue (Figure 1C) and BALF (Figure 1D) were also significantly elevated in B+I mice compared to burn or sham mice 24 h post-injury. Together, these data indicate that, similar to that observed in humans, the model of combined B+I injury described here exhibits more basal mortality, lung damage, and influx of cells in the lung when compared to burn injury alone.

### 2.2. Immune Cells within the Lung Are Differentially Impacted by Burn and B+I Injury Early and Late after Injury

We and others reported that burn or inhalation injury alone results in an acute accumulation of immune cells into the lung cavity, whereby these cells have increased antimicrobial activity [30]. In order to characterize the immune cell populations within the lung tissue and airway space, cells were isolated from the lung tissue and BALF of mice that had undergone sham, burn, or combined injury. Cells were collected 24 h (Figure 2A) or 14 days after injury (Figure 2B) and examined for macrophage or neutrophil markers via flow cytometric analysis. At 24 h post-injury, the number of neutrophils and macrophages in the BALF from B+I mice were significantly higher than those within the BALF from burn injury alone. At 14 days post-injury, the number of neutrophils in BALF from B+I-injured mice were significantly higher than burn injury alone. In addition, at this later time point, we saw a significantly higher number of macrophages in the BALF following B+I injury compared to that of the BALF from burn- *or* sham-injured mice. These data demonstrate that combined burn and smoke inhalation (B+I) injury leads to elevated levels of immune cells in the airway. 

### 2.3. Lung and Soluble Immune Mediators in the Lung and Peripheral Blood Are Differentially Impacted by Burn and B+I Injury Early after Injury

To further characterize the immune response in the lung microenvironment 24 h post-injury, BALF was analyzed on a Bio-Rad 33-plex array to identify cytokines that had been secreted into the airways. We found significant increases in the levels of the following cytokines in the airways of B+I animals compared to sham or burn alone: IL-1β, IL-6, IL-16, CCL7, CCL17, and CCL24 (Figure 3A). Our group demonstrated that burn or inhalation injury alone also promotes a systemic immune dysfunction which correlates with poor outcomes in burn and inhalation injury patients [41,43,44]. Thus, to evaluate the systemic immune response to B+I injury, we utilized an 8-plex cytokine detection panel and observed that IL-6 was significantly elevated in the plasma of B+I mice when compared to burn injury or sham injury, whereas MCP-1 was elevated in B+I mice compared to sham injury mice only, as burn injury alone also induced significant levels of MCP1 (Figure 3B). Both IL-6 and MCP-1 correlate strongly with a poor outcome in humans. Together, these data indicate that B+I injury results in an exacerbated local and systemic immune response when compared to burn injury alone.

### 2.4. Immune Gene Transcriptome Analysis Revealed Pulmonary and Systemic Differential Immune Gene Expression in B+I Injury

To examine the expression of genes and immune pathways that are triggered as a result of B+I injury, we employed NanoString technology [45]. We utilized the commercially available mouse Cancer Immune Panel, which allows for 770 mRNA targets to be quantified simultaneously, spanning genes covering both the adaptive and innate immune response and common immunometabolic genes. To evaluate acute changes in immune gene expression within the lung microenvironment, we isolated mRNA from whole lung 24 h after sham, burn, or B+I injury. Volcano plots in Figure 4 demonstrate the change in gene expression within the lung tissue along with its associated significance following burn (Figure 4A) or B+I (Figure 4B) injury relative to sham injury, as well as following B+I injury relative to burn injury (Figure 4C). Significantly altered genes (*p* < 0.05) are presented in Appendix A. Fifteen genes were found to be significantly altered by burn injury alone compared to sham mice (Figure 4A, 4 genes downregulated, 11 genes upregulated). In contrast, only three genes were significantly altered by B+I injury when compared to gene expression in sham mice (Figure 4B, two genes downregulated, one gene upregulated). Notably, expression of CD22 and granzyme A (Gzma) were downregulated in both B+I and burn mice (Figure 4A–C). In addition, 29 genes were significantly altered when comparing B+I vs. burn injury mice (Figure 4D, 15 genes downregulated, 14 genes upregulated). 

To elucidate differences in pulmonary immune signaling pathways affected by B+I injury vs. burn alone, we performed a focused analysis on canonical immune pathways utilizing these NanoString data and the Ingenuity Pathway Analysis (IPA; Figure 4E). Here, we determined that based on the Z-scores, B+I significantly altered the activity of seven pathways compared to burn injury alone, with significant increased activation of the acute-phase response pathways, NOS production, IL-6 signaling, and general “wound healing” pathways. 

We also evaluated the expression of genes and immune pathways within the periphery that are triggered by B+I injury. Here, we isolated mRNA from the spleen 24 h after injury. Volcano plots in Figure 5 demonstrate the change in gene expression along with its associated significance, following burn (Figure 5A) or B+I (Figure 5B) injury relative to sham injury, and following B+I injury relative to burn injury (Figure 5C). Significantly altered genes (*p* < 0.05) are presented in Appendix A. In total, 34 genes were significantly altered by burn injury only compared to sham-injured mice (Figure 5A, 15 genes downregulated, 19 genes upregulated). In addition, only six genes were significantly altered by B+I injury when compared to gene expression in sham-injured mice (Figure 5B, two genes downregulated, four gene upregulated). Finally, 17 genes were significantly altered when comparing B+I injury vs. burn injury alone (Figure 5C, 11 genes downregulated, 6 genes upregulated). Interestingly, IPA analysis did not reveal any significant effects on systemic immune signaling pathways when B+I was compared to burn injury alone. Figure 5D summarizes the number and identity of genes that are impacted in a unique or common fashion to each injury type (when quantified vs. sham). 

We also evaluated the effect of the local (lung) and systemic (spleen) microenvironments on gene expression and pathway activation. Figure 5E,F summarizes the number and identity of genes that are impacted in a unique or common fashion within each microenvironment (lung or spleen) following burn injury alone (Figure 5E, quantified vs. sham) or B+I injury (Figure 5F, quantified vs. burn alone). These data revealed that burn injury has quite discrete effects on the lung and spleen, with only one gene difference common to both. 

Taken together, these data indicate that B+I injury induces both unique and overlapping gene and immune signaling pathways when compared to burn injury alone. Not surprisingly the largest effect is seen within the local environment of the lung. 

### 2.5. mTOR Regulates Systemic and Local Hyper-Responses to Burn and B+I Injury

It is well accepted that the mTOR axis regulates inflammation and innate immune cell function [46,47,48]; however, no study has established a mechanistic link between the immune dysfunction, subsequent infection control, and the mTOR axis after B+I injury. We hypothesized that the B+I injury model would allow us to investigate the impact of in vivo pharmacological mTOR inhibition on pulmonary and systemic immune transcriptomes, including associated molecular mechanisms of bacterial susceptibility. 

Specifically, we utilized systemic administration of rapamycin (daily, starting 7 days before the injury) to inhibit mTOR activity at the time of burn or B+I injury. We and others published that a 7-day protocol of high-dose rapamycin (4–10 mg/kg/day) inhibits intracellular phosphorylation of the downstream molecule S6 via flow cytometry after burn injury in immune cells [41,46,49,50]. As we previously reported, we did not observe a significant difference in mortality between vehicle-treated burn-only and rapamycin-treated burn-only mice (data not shown [41]). Conversely, we observed that rapamycin significantly decreased the survival of B+I mice compared to B+I which received the vehicle. Specifically, the survival rate was reduced to ~25% over the first 48 h after injury (Figure 6A). In order to investigate mechanisms associated with this change in mortality, BALF, plasma, whole lung, and spleen were harvested 24 h after injury and the immune repertoire evaluated. 

Rapamycin-treated B+I-injured mice presented with a significantly altered cytokine profile in the plasma with higher levels of IL-6 and IFNγ, and reduced levels of TNFα and IL-2 in B+I plasma compared to vehicle-treated B+I-injured mice (Figure 6B). In contrast, within the lung microenvironment, we observed significantly reduced IL-6 expression in the BALF of rapamycin-treated B+I mice when compared to vehicle-treated controls. Taken together, these data demonstrate a role for mTOR in regulating systemic and local immune dysfunction, with differential effects in the lung and in the periphery, after combined B+I injury. 

In order to unveil which genes and pathways were under the control of mTOR, we again turned to NanoString analysis. Mice were administered either rapamycin or a vehicle for 7 days before receiving either B+I, burn, or sham injury. mRNA was extracted from whole lung and spleen 24 h after injury (Appendix A), and the immune transcriptome was analyzed as above. Volcano plots in Figure 7 demonstrate the change in gene expression along with its associated significance following B+I injury relative to vehicle-treated controls, with the associated statistical significance of each gene. 

These data reveal that following B+I injury, the mTOR blockade at the time of injury by rapamycin induced significant expression changes in 41 lung immune genes (8 upregulated, 33 downregulated; Figure 7A). For the spleen, representing the peripheral immune system, rapamycin induced significant expression changes in 41 spleen immune genes (11 upregulated, 25 downregulated; Figure 7B). In each case, significantly altered gene expression (*p* < 0.01) data are presented in Appendix A. Z-scores of canonical pathways significantly (*p* < 0.05) altered with rapamycin treatment in B+I mice compared to vehicle-treated mice in spleen and lung are also reported in Figure 7; in the lung, rapamycin induces a significant downregulation of IL-6, natural killer (NK) cells, and wound healing-associated pathways. In the spleen, rapamycin significantly impairs pathways involved in NO/ROS production in macrophages, wound healing, and leukocyte extravasation. Together, these data demonstrate that the blockade of mTOR at time of injury with rapamycin has significant effects, but of differing identity between compartments, on immune signaling pathways in both the local and systemic environments with B+I injury. 

### 2.6. B+I Injury, Compared to Sham Injury, Increases Susceptibility to Bacterial Infections Early and Late after Injury; Susceptibility Early and Late after Injury Is Differentially Regulated by mTOR

We previously demonstrated that mTOR activation is observed in pulmonary neutrophils and macrophages early after cutaneous burn injury and that inhibition of mTOR at the time of injury with rapamycin induces chronic bacterial susceptibility compared to vehicle-treated burn-injured controls [41]. Thus, 14 days after injury, we inoculated surviving B+I or sham-injured mice that had received rapamycin or the vehicle control with *Pseudomonas aeruginosa* (strain PAO1, a clinically relevant opportunistic pathogen that frequently causes fatal pneumonia in burn patients). Then, 24 h after inoculation, the spleen and lung were harvested to quantify bacterial infection and organ dissemination using culture techniques. We observed significantly greater numbers of PAO1 bacteria recovered from the spleen and lungs of B+I mice that did not receive rapamycin when compared to sham-injured mice. However, in the rapamycin pretreated mice we harvested significantly fewer numbers of PAO1 from the spleen and lungs of B+I mice, which were pretreated with rapamycin when compared to sham-injured mice that also received rapamycin. Finally, significantly lower numbers of PAO1 were observed in B+I-injured mice pretreated with rapamycin when compared to B+I mice that received the vehicle pretreatment (Figure 8). 

Taken together, these results suggest that the B+I model recapitulates the clinical presentation of combined injury in humans with significant bacterial susceptibility, including the development of lung infections both early and late after injury. In addition, the mTOR blockade at the time of injury increases the ability of mice to clear infection late after injury. 

## 3. Discussion

A significant challenge in the application of translational research to improve outcomes to burn and inhalation injury has been that existing models demonstrated chronic (~14 days post-injury) and significantly enhanced hyper-innate immune responses [30,41,51,52] rather than the accepted clinical picture of general immunosuppression. We showed that models require repeated exposure to bacterial insult for burn-mediated immunosuppression to become apparent [30] in “single hit” burn models, which is not dissimilar to clinical observations. This is likely due to the long-lived, burn-induced upregulation (“burn-priming”) of the innate arm of the immune system [53,54], which is sufficient to protect patients against a single bacterial insult but which collapses with subsequent challenge. We and others showed that there is an increased neutrophil presence in the lung vasculature early and late after burn injury [55,56,57], which contributes to the improved infection outcome in burn mice following single bacterial challenge. 

This study investigated the effect of combined woodsmoke inhalation, a common comorbidity with burn injury, on the immune priming induced by cutaneous burn injury. Our data indicate that our B+I model recapitulates aspects of immune dysregulation observed in patients who are admitted to the hospital due to burn and inhalation injury, and is characterized by increased mortality, lung damage, innate immune cell recruitment to the airway, and susceptibility to infection [58,59]. We found that B+I injury leads to higher mortality and immune dysfunction than burn alone. For instance, we found that multiple cytokines are secreted into the airway after injury, including immunomodulatory cytokines such as IL-1β, IL-6, and IL-16 (Figure 9A), at elevated levels compared to burn-injured mice. Similarly, we found increased proinflammatory cytokines and chemokines such as elevated IL-6 and MCP-1 in the plasma. These cytokines are commonly associated with shock and the associated SIRS phase found to be elevated in patients who are being treated for burn injury [9,60]. Studies have indicated that elevated and sustained peripheral levels of IL-1β and IL-6 correlate with poor clinical outcomes among patients [61]. Other chemokines that we found to be elevated in the BALF following B+I are involved in the recruitment of T cells, monocytes, macrophages, dendritic cells, and neutrophils. Recruitment of these cells leads to increased recruitment and activation of immune cells at the site of injury, potentially leading to worsened damage of the lungs. When challenged with bacteria commonly associated with pneumonia in burn patients, B+I injury compared to sham injury increases susceptibility to systemic and pulmonary bacterial colonization late after injury (Figure 9B). Thus, this novel murine model allows us to uniquely examine alterations in the lung tissue, as this is an organ that cannot be utilized for study in humans. 

We examined the immune gene transcriptome using NanoString analysis to identify pulmonary and systemic immune gene expression and pathway activity in burn and B+I injuries. There were clear commonalities and differences between burn and B+I injuries in the lung and spleen tissues; for example, CD22 and granzyme A (Gzma) were similarly downregulated in both B+I and burn mice vs. sham-injured mice. These expression data also led to the discovery of multiple immune genes and associated canonical immune pathways that were significantly affected by the combination of inhalation injury and burn injury compared to burn alone, with significant increases in acute-phase response pathways, NOS production, IL-6 signaling, and wound healing pathways. Taken together, these are examples of potential mechanisms, therapeutic targets, and biomarkers within specific tissue types. Indeed, our data revealed that burn has discrete effects on the lung and spleen, with only Plaur1 being commonly induced in both compartments Similarly, B+I injury acutely impacts genes unique to the spleen or lung, with only Plaur1, again, common to both microenvironments. 

It is well accepted that the mTOR axis regulates inflammation and innate immune cell function [46,47,48], and we hypothesized that mTOR regulates systemic and local hyper-responses to burn and B+I injury and, thus, influences susceptibility to bacterial infections. Specifically, burn and B+I injury-induced DAMPs activate innate immune (e.g., macrophages and neutrophils) and stromal cells (e.g., pulmonary epithelium) through PRR and activation of the mTOR/PPARγ axis [58,59]. mTOR activation drives the execution of metabolic cellular programing and inflammatory functions [46] as well as the upregulation of the negative regulator of inflammation, PPARγ, which, in turn, represses overactive immune responses [39]. PPARγ also regulates cellular metabolic reprogramming (i.e., switch from respiration to glycolysis), which directly affects innate effector function. Although previous studies have explored immunological dysfunction during burn injury, no study has established a mechanistic link between the immune dysfunction, subsequent infection control, and the mTOR axis after B+I injury.

In this study we hypothesized that the B+I injury model would allow us to investigate the impact of in vivo pharmacological mTOR inhibition on pulmonary and systemic immune transcriptomes including associated molecular mechanisms of bacterial susceptibility. Rapamycin significantly decreased the survival of B+I mice, reducing the survival rate from ~50% to ~25%. At 24 h after injury, we observed that rapamycin significantly elevated plasma IL-6 and IFNγ in B+I plasma compared to vehicle-treated controls, representing amplified SIRS (Figure 9B). In contrast, we observed a significant rapamycin-dependent reduction in pulmonary IL-6 and IL-2. Immune transcriptome analysis of these surviving mice at 24 h after injury highlighted common and unique gene expression changes. Thus, although not directly evaluated here, these data suggest B+I-induced perturbations in pulmonary NK cell, T cell, and neutrophil effector functions. In the systemic compartment, we also observed significant rapamycin-dependent gene expression with a notable increase in expression of the interleukin-6 cytokine family signal transducer (IL6ST), which indicates increased IL-6 expression and action [62], and dedicator Of cytokinesis 9 (DOCK9), an emerging master-regulator of cytokine response to PRR responses [63]. The net effect of these expression changes between rapamycin-treated injured mice and vehicle control mice were revealed by IPA and pointed towards a significant reduction in pulmonary IL-6 and NK cell signaling, and a general reduction in the pathway associated with tissue healing, with the loss of mTOR signaling at the time of injury. These data agree with the cytokine data in the BALF that mTOR inhibition does indeed reduce the inflammatory response in the lung after injury. Systematically, we see less of an impact on the IL-6 pathway in the spleen, although we do see an increase in IL-6 in the serum with rapamycin inhibition at the time of injury and a high level of mortality.

We also demonstrated that B+I injury, compared to sham injury, increases susceptibility to bacterial infections late after injury and our data here indicate that this susceptibility is also regulated by mTOR. mTOR inhibition at the time of injury in the B+I model prevented chronic (14-day) systemic and pulmonary infection with PAO1 (Figure 9B). We suspect this is due to a lack of pulmonary damage and immune dysfunction early after injury. More specifically, we hypothesize that rapamycin treatment interrupts the vicious cycle of inflammation and ineffective immune dysfunction seen after injury by reducing the early pulmonary proinflammatory response. 

A limitation to this study is that we only evaluated the mTOR-dependent responses in mice surviving a heightened systemic shock period, and the amplitude of cytokine response in the nonsurvivors is likely even higher than the survivors. We also only investigated systemic inhibition of mTOR; however, we are currently investigating the role of cell-specific mTOR in these responses. Regardless, we uncovered a series of immune genes that are crucial to survival after injury and to bacterial resistance, which arise in a tissue-specific manner and have the potential to be used as therapeutic targets. Although we are not proposing that pretreatment with rapamycin before injury is a viable clinical treatment, it revealed how important it is to study tissue-specific responses and sets the stage for informing the timing of interventions aimed at breaking these cycles of injury- and immune-induced tissue damage. It is also tempting to speculate that uncoupling the naturally occurring mTOR/PPARγ axis which acts to transduce, amplify, and then dampen down inflammatory responses could provide therapeutic efficacy. 

## 4. Materials and Methods

### 4.1. Murine Burn and Inhalation Injury Models

Eight- to twelve-week-old female C57BL/6 mice weighing between 18–21 g were purchased from Taconic Farms for use in this study. Mice were anesthetized using tribromoethanol (475 mg/kg body weight, Sigma-Aldrich, Burlington, MA, USA) and then shaved dorsally and given a subcutaneous injection of morphine (3 mg/kg body weight, West-Ward Pharmaceuticals, Eatontown, NJ, USA). A subset of mice were administered a 20% total body surface area burn injury as previously described [64,65,66]. Briefly, a 65 g copper rod (1.9 cm in diameter) was heated to 100 °C in a hot water bath and then held to the dorsum/flank of the animal for four separate applications lasting 10 s each to achieve a full-thickness contact burn. Following burn or sham procedures, all mice were placed on an intubation platform (Penn Century) and intubated with a laryngoscope-guided catheter (22Gx1”, Exel) inserted in the trachea between the vocal cords. Inhalation injury was then applied to a subset of mice as previously described [43]. Intubated mice were secured to a platform and placed into an animal induction chamber (Stoelting NC9296517) that receives air flow from an adjoining side-arm flask. To generate smoke, 50g of sectioned plywood (2.5 cm × 8 cm, Lowe’s item #12206, model #776391100000) was added to the flask and heated to 500 °C via hotplate air pumped into the flask at constant pressure, which enabled a constant flow of smoke into the induction chamber. Smoke density in the chamber was visually assessed and considered appropriate when visual obstruction was obtained at a depth of 1–1.5 inches from the chamber wall. The chamber door was vented as needed to maintain a consistent thickness of smoke. Each animal received three cycles of a two-minute smoke exposure, followed by a one-minute clean air break (fully removing the mouse from the chamber). Animals were allowed to recover on a heated surface until anesthesia subsided and then were given an intraperitoneal injection of lactated Ringer’s solution (0.1 mL/g body weight; Hospira, Lake Forest, IL, USA). Mice were then placed in individual cages, given food and morphine water ad libitum, and monitored twice daily. Sham animals underwent all interventions with the exceptions of copper rod application or smoke exposure within the induction chamber. All animals were housed in the UNC Chapel Hill Department of Comparative Medicine’s specific pathogen-free animal housing facilities and all protocols and procedures were approved by the University of North Carolina’s Institutional Animal Care and Use Committee in accordance with NIH-specified guidelines. 

### 4.2. Peripheral Blood Plasma and Bronchoalveolar Lavage Collection

Mice were anesthetized by isoflurane vapor exposure before immediately undergoing retro-orbital bleed, followed immediately by cervical dislocation. Approximately 1 mL of blood was collected in a 1.5 mL Eppendorf tube containing 100 μL of 0.5 M EDTA and centrifuged (5000× *g* for 15 min). The clear plasma layer was carefully pipetted into new Eppendorf tubes for storage at −80 °C. Simultaneously, bronchoalveolar lavage (BALF) was conducted on the mouse to collect cells and protein from the airway space as previously described [67,68,69]. Dissection was performed to expose the trachea and a catheter (22G × 1”, Exel) was inserted into the trachea such that a syringe containing 1 mL of PBS was attached to the end of the catheter for sequential washes. For each 1 mL of PBS injected into the airway space, 0.7 mL of BALF fluid was collected. Each mouse received a total of three BALF washes. New syringes and PBS were used for each wash while the inserted catheter remained the same. BALF fluid from each mouse was pooled, centrifuged (5 min, 12,000 RCF), and the resulting layers partitioned. Supernatants and cell pellets were collected and stored (−20 °C) for further analysis at a later date. Total protein content of BALF supernatant was quantified using a Bradford colorimetric protein assay according to the manufacturer’s instructions (Bio-Rad, Hercules, CA, USA, #5000006).

### 4.3. Lung and Spleen Tissue Isolation and Processing

Following BALF, whole lung and spleen tissue were surgically removed and minced using sterile razor blades. Finely diced lung tissue was then placed in 4 mL of PBS + 10% fetal bovine serum (PBS + FBS) supplemented with 0.1 µg/mouse DNase and 1500 μ/mouse collagenase. Samples were then shaken at 250 rpm at 37 °C for 1h to digest tissue and yield a single-cell suspension as previously described [64]. Digested lung solution was then filtered with a 100 µm cell strainer and pelleted (5 min, 300 g). ACK lysis buffer (2 min exposure) was used to remove red blood cells present in the pellet before washing and resuspending in PBS+FBS for storage until future use. Cells obtained from BALF washes and whole lung tissue processing were counted using a hemocytometer with 0.01% trypan blue viability dye. 

### 4.4. Cytokine and Chemokine Assay

Samples were processed for analysis on a Bio-Plex mouse chemokine 33-plex panel (Bio-Rad #12002231) or Bio-Plex Pro Mouse Cytokine Group 1 7-plex panel (Bio-Rad, #L6000004C6) according to the manufacturer’s protocol. Samples were washed using the Bio-Plex Pro wash station (Bio-Rad #30034376) and analyzed using the Bio-Plex MAGPIX Multiplex reader (Bio-Rad #171015001). 

### 4.5. Flow Cytometry

Fc receptors on cells were blocked using anti-mouse CD16/32 ( BD Biosciences, San Jose, CA, USA). Antibodies against CD45, CD11c, CD11b, Ly6G, and F4/80 were then utilized to stain cells. Cells were then washed twice and fixed with 1% paraformaldehyde. Stained samples were examined on a Dako CyAn (Beckman Coulter) and data obtained were analyzed using Summit software (Beckman Coulter). Cells identified as CD45+ were than examined to identify neutrophils (CD45+CD11b+CD11c-Ly6G+) and macrophages (CD45+CD11c+Ly6G-) as previously described [64,65]. 

### 4.6. Immune Gene Detection and Quantification

Isolation of mRNA was performed as described previously [70]. Briefly, cells were lysed with TRIZOL buffer (Sigma) and total RNA was isolated by chloroform extraction and quantified using a nanodrop 2000^TM^ spectrophotometer. NanoString technology and the nCounter Mouse PanCancer Immune Panel was used to simultaneously evaluate 770 immune and metabolic-related mRNAs in each sample [45]. Each sample was run in triplicate. Briefly, a total of 100 ng of mRNA was hybridized to report-capture probe pairs (CodeSets) at 65 °C for 18 h. After this solution-phase hybridization, the nCounter Prep Station was used to remove excess probes, align the probe/target complexes, and immobilize these complexes in the nCounter cartridge. The nCounter cartridge was then placed in a digital analyzer for image acquisition and data processing. Color codes designating mRNA targets of interest were directly imaged on the surface of the cartridge. The expression level of each gene was measured by counting the number of times the color-coded barcode for that gene was detected, and the barcode counts tabulated. nSolver v4.0, an integrated analysis platform, was used to generate appropriate data normalization as well as fold changes, resulting in ratios and differential expression. The nCounter™ v4.0 Advanced Analysis and Ingenuity Pathway Analysis, along with robust R statistics, was used to identify pathway-specific responses [45].

### 4.7. Rapamycin Blockade of mTOR

Rapamycin (LC Laboratories, Woburn, MA; 4 mg/kg/day) or a vehicle control was administered daily, starting 7 days before the burn injury via an intraperitoneal injection, as previously described [41]. 

### 4.8. Infection with Pseudomonas aeruginosa

*Pseudomonas aeruginosa* (PAO1) from 10% glycerol stock was swabbed into LB and allowed to grow on a shaker at 37 °C overnight. The following day, 500 μL of overnight culture was added to 4mL of LB and placed on the shaker at 37 °C to achieve log-phase growth. The OD_600_ was measured with a photospectrometer (Eppendorf, No. 6131 24465, Enfield, CT, USA) and used to calculate total CFUs and dilute to 2 × 10^6^ CFU PAO1 per mouse. The diluted PAO1 culture was centrifuged (14,000 rpm × 2 min) and washed with 1% proteose-peptone in PBS (PP-PBS) twice before resuspending in PBS for *i/v* (tail vein) -administered infection 14 days after BI or sham injury. Then, 24 h after infection, mice were euthanized and the lungs and spleen were collected, weighed, and homogenized in 750 μL of PBS. Each solution was serially diluted, streaked onto LB agar plates, and allowed to grow overnight at 37 °C. CFUs were counted the following day and used to back-calculate CFUs per g tissue.

### 4.9. Statistical Analysis

Analysis was conducted after data normality was met using the method of D’Agostino and Pearson. All data were examined in GraphPad Prism Version 9.0 for Windows and analyzed using a one-way analysis of variance (ANOVA) with a Tukey post-test. For NanoString, a negative binomial mixture model, simplified negative binomial model, or loglinear model was used depending on each gene’s mean expression compared to the background threshold, as we described previously [71,72]. Multiple testing correction was performed using the method of Benjamini–Yekutieli. Causal network analysis was performed using IPA. Data are displayed as mean +/− standard error of the mean (SEM). Statistical significance is indicated as * *p* < 0.05, ** *p* < 0.005, and *** *p* < 0.001. Where indicated, data represent a representative experiment (from three independent experiments) performed in a single run. Experimental data were not pooled in order to be presented as a bar graph with SEM bars due to the interexperiment variability. 

## Figures and Tables

**Figure 1 ijms-23-08779-f001:**
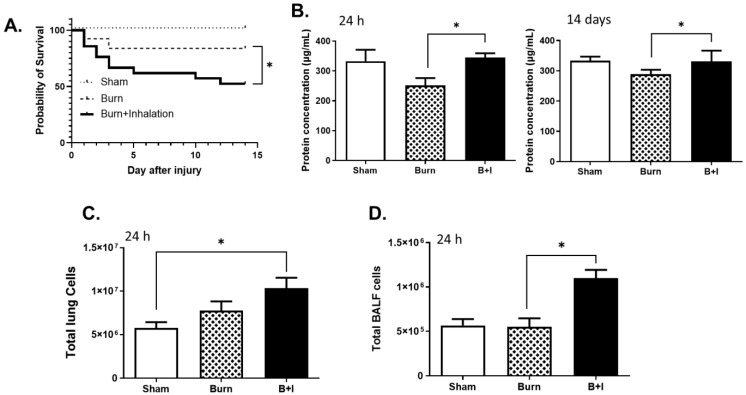
Combined burn and smoke inhalation (B+I) injury leads to higher mortality than burn injury alone, with elevated levels of cells and protein within the lung tissue and airway. Mice underwent either B+I, burn, or sham injury. Lung tissue or BALF was harvested at 24 h or 14 days after injury. (**A**) Survival of mice from injury to 14 days after injury was evaluated (n = 12 per group); (**B**) total proteins in bronchoalveolar lavage fluid (BALF) were assayed by BCA (n = 6 per group); (**C**,**D**) total immune cell numbers within the total lung tissue (**C**) and BALF (**D**) were evaluated at 24 h post-injury (n = 6 per group) by flow cytometry. In each case, data are presented +/− SEM, with significance represented as * *p* < 0.05. Data are representative of three independent experiments.

**Figure 2 ijms-23-08779-f002:**
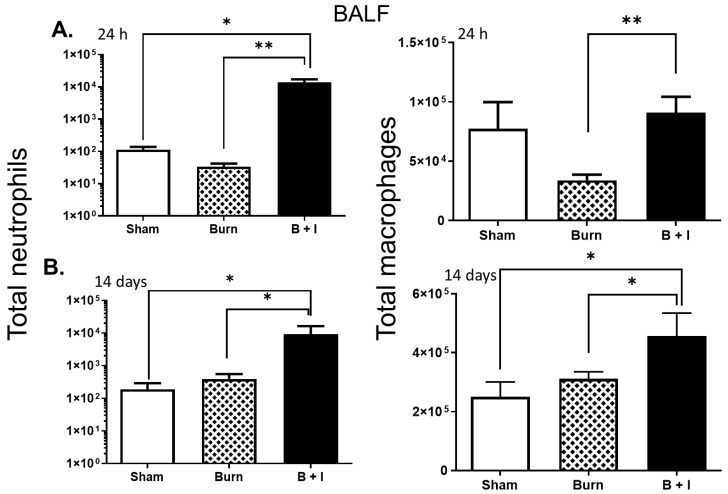
Innate immune cells are recruited to the airspace and to the pulmonary tissue in an injury-dependent manner early and late after injury. Mice underwent sham, burn, or combined injury, and cells isolated were examined for macrophage or neutrophil markers via flow cytometric analysis 24 h (**A**) or 14 days (**B**) after injury (n = 6 mice per group). Neutrophil (left panel) and macrophage (right panel) numbers were determined within the BALF by flow cytometry. In each case, data are presented +/− SEM, with significance represented as * *p* < 0.05, ** *p* < 0.01. These data are representative of three independent experiments.

**Figure 3 ijms-23-08779-f003:**
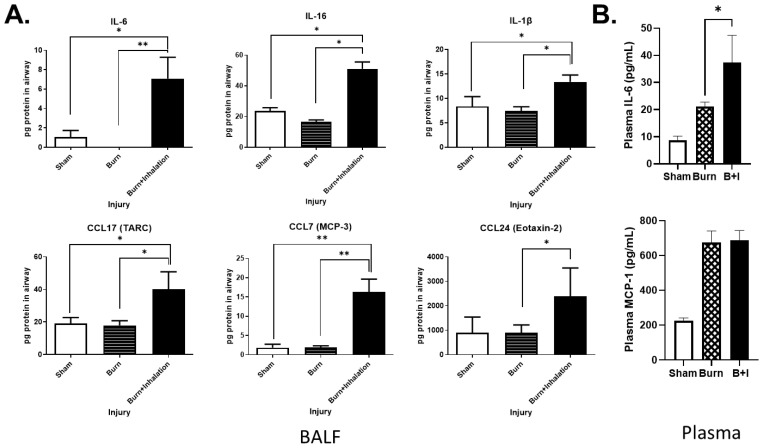
B+I injury significantly alters BALF and plasma cytokines and chemokines compared to burn injury or sham injury alone. Mice underwent sham, burn, or combined injury, and were sacrificed at 24 h post-injury (n = 6 per group). (**A**) Supernatant collected from BALF washes was analyzed using a Bio-Plex mouse cytokine and chemokine 33-plex panel. (**B**) Plasma was analyzed using a Bio-Plex mouse cytokine 8-plex panel. Data shown are +/−SEM; * *p* < 0.05, ** *p* < 0.01.

**Figure 4 ijms-23-08779-f004:**
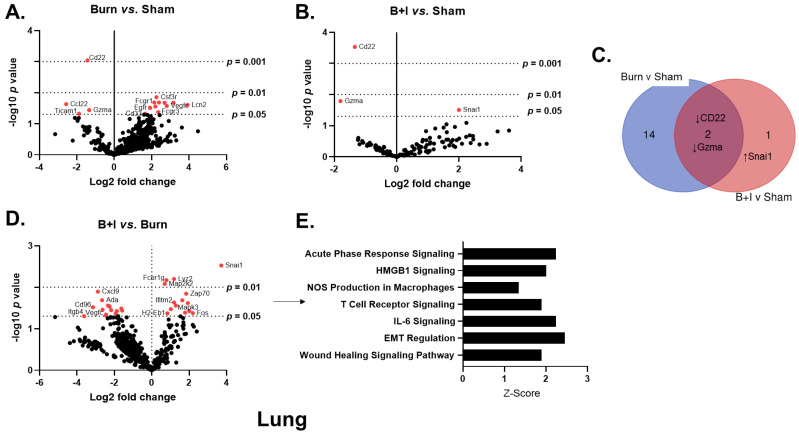
Injury type induces differential pulmonary gene expression and associated canonical pathway activity. Mice underwent sham, burn, or combined B+I injury, and were sacrificed at 24 h post-injury (n = 6 per group). mRNA was extracted from whole lung tissue and NanoString analysis performed with a mouse Cancer Immune Panel, allowing expression analysis of 770 immune and metabolic genes. (**A**–**C**) are volcano plots of injury-induced differential gene expression, where red dots signify *p* < 0.05 (−log10(*p*-value) > 1.301). A Venn diagram was constructed (**C**) to summarize the number and identity of genes that are impacted in a unique or common fashion to each injury type (when quantified vs. sham). Differential gene expression and pathway Z-scores between B+I and burn alone were analyzed via Ingenuity Pathway Analysis (**D**,**E**); only significantly altered (*p* < 0.5) pathways are shown.

**Figure 5 ijms-23-08779-f005:**
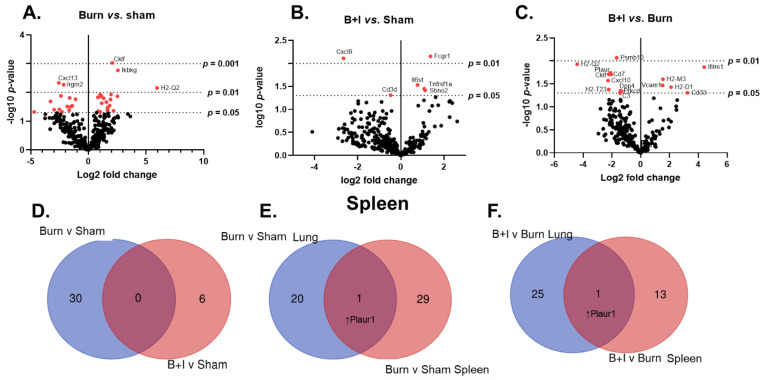
Injury type induces differential spleen (systemic) gene expression. Mice underwent sham, burn, or combined B+I injury, and were sacrificed 24 h post-injury (n = 6 per group). mRNA was extracted from whole lung tissue and NanoString analysis, allowing expression analysis of 770 immune and metabolic genes. (**A**–**C**) Volcano plots of injury-induced differential gene expression, where red dots signify *p* < 0.05 (−log10(*p*-value) > 1.301). (**D**–**F**) Venn diagrams summarize the number and identity of genes that are impacted in a unique or common fashion to each (**D**) injury type (when quantified vs. sham), (**E**,**F**) organ studied after (**E**) burn injury alone (when quantified vs. sham) or (**F**) B+I injury (when quantified vs. burn alone).

**Figure 6 ijms-23-08779-f006:**
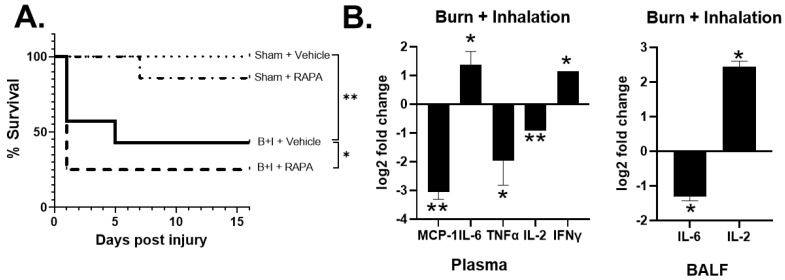
Rapamycin-induced significant mortality after B+I injury with altered local and systemic cytokine profiles. Mice were administered either rapamycin or vehicle for 7 days before receiving either B+I or sham injury (n = 12 per group). (**A**) Survival of mice from injury to 14 days after injury was evaluated. (**B**) Plasma and BALF (n = 6 per group) were harvested 24 h after injury and analyzed using a Bio-Plex mouse cytokine panel 8-plex assay; log2 fold change is shown vs. vehicle-treated injured control mice. In each case, data are presented +/− SEM, with significance represented as * *p* < 0.05, ** *p* < 0.01. These data are representative of three independent experiments.

**Figure 7 ijms-23-08779-f007:**
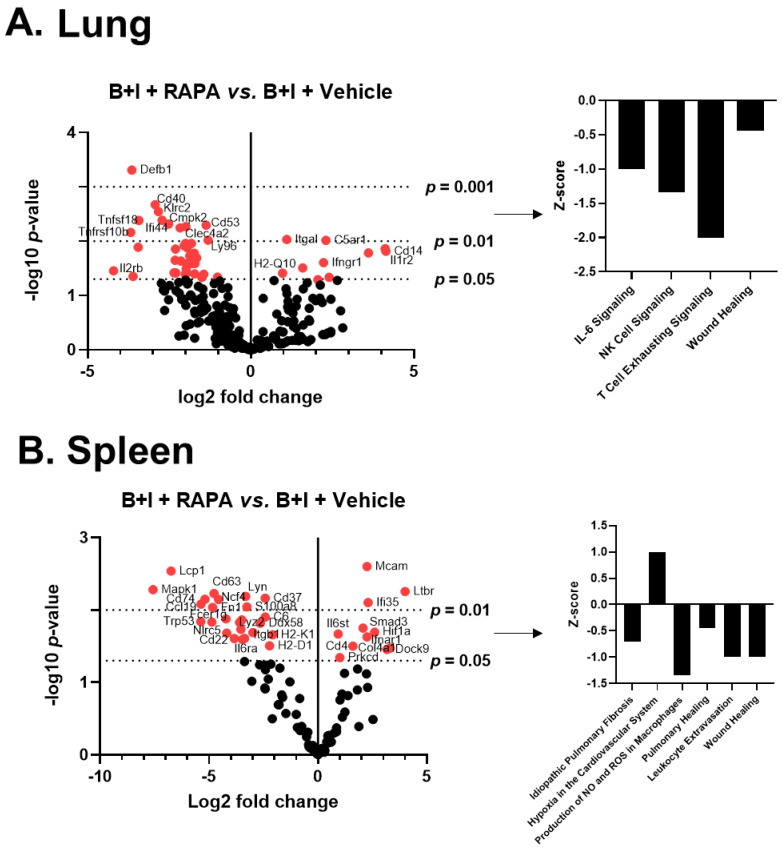
mTOR inhibition via rapamycin at the time of injury induced early differential gene expression and canonical pathway activity in B+I mice compared to vehicle controls. Mice were administered either rapamycin or vehicle pretreatment for 7 days before receiving B+I injury (n = 6 per group). mRNA was extracted from (**A**) whole lung and (**B**) spleen tissue and run on NanoString. Differential gene expression and pathway Z-scores were analyzed via nSolver and Ingenuity Pathway Analysis, respectively. Volcano plots are of rapamycin-induced differential lung gene expression after B+I + rapamycin (“RAPA”) vs. B+I + vehicle controls, where red dots signify *p* < 0.05 (–log10(*p*-value) > 1.301). For the Z-plots, only significantly altered (*p* < 0.5) pathways are shown.

**Figure 8 ijms-23-08779-f008:**
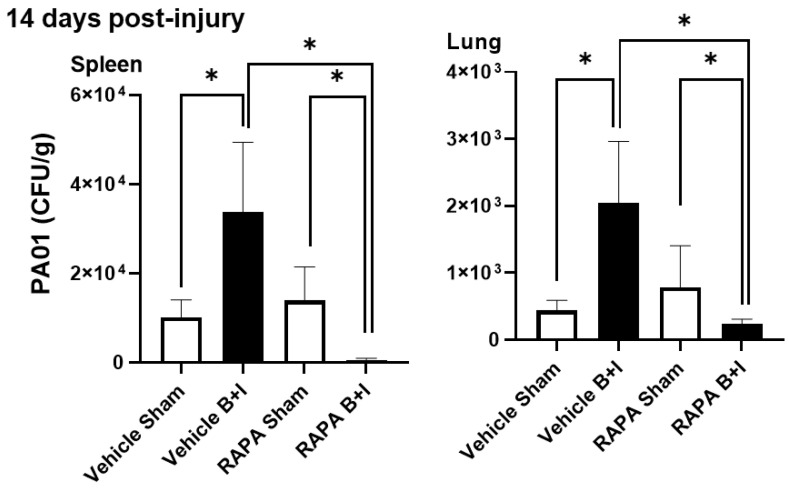
B+I injury, compared to sham injury, increases susceptibility to bacterial infections late after injury; susceptibility early and late after injury is differentially regulated by mTOR. Mice were administered either rapamycin or vehicle pretreatment for 7 days before receiving either B+I or sham injury (n = 12 per group). Surviving mice were *i/v* inoculated with *Pseudomonas aeruginosa* (strain PA01) 14 days after B+I or sham injury (n = 6 per group). After 24 h, spleen and lung were harvested and plated on agar to assess bacterial infection, and colony forming units (CFUs) calculated for each injury/treatment condition. In each case, data are presented +/− SEM, with significance represented as * *p* < 0.05. These data are representative of three experiments.

**Figure 9 ijms-23-08779-f009:**
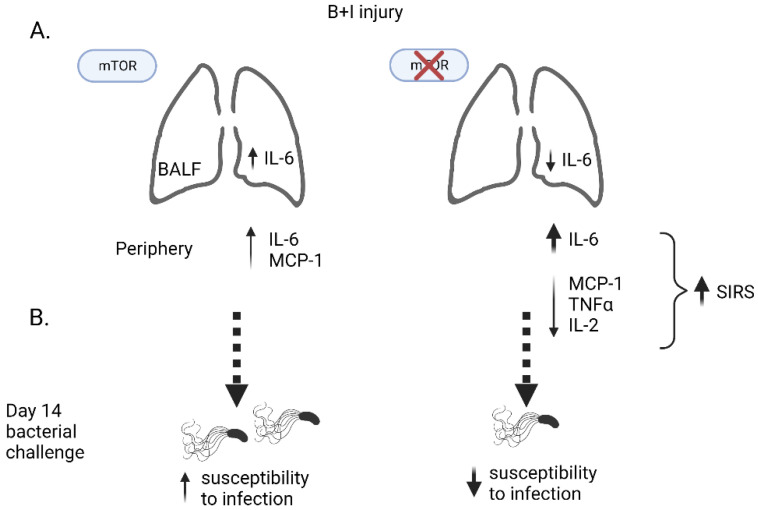
B+I injury, compared to sham injury, induces increased cytokine expression in the lung and plasma (**A**, left), and increases susceptibility to bacterial infections late after injury (**B**, left). These cytokine effects in the lung and plasma are differentially regulated by mTOR inhibition at the time of injury (**A**, right), with enhanced SIRS (systemic inflammatory response syndrome). Infection susceptibility late after injury is ameliorated by mTOR inhibition at time of injury (**B**, right).

## Data Availability

The data presented in this study are openly available in the Appendix A.

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
