# Peer review of "Characterization of the Basal and mTOR-Dependent Acute Pulmonary and Systemic Immune Response in a Murine Model of Combined Burn and Inhalation Injury"

_ijms, 2022, doi:10.3390/ijms23158779_

Round 1
Reviewer 1 Report
The study is very thorough and has built a murine model of burn+ inhalation injury to investigate infection susceptibility. They demonstrate the increased characteristics of infection and mortality by burn and inhalation combined, versus burn or inhalation alone. They also show a mechanism by which this can occur by the investigation on mTOR signaling upon burn and inhalation. They also show do gene expression profiling to explore for immune signaling pathways and genes that could function in this situation giving therapeutic targets for investigation and drug discovery.
Some further comments:
1. Introduction (lines 50 and 51): It will be useful to define ‘SIRS’ and ‘CARS’.
2. Figure 1: Shows an increase in protein concentration in the lung BALF after inhalation. How would you describe the significance of this?
3. The inhalation injury only component has been shown only in figures 1, 2 and 4. They show a tendency of interaction between the burn effect and the inhalation injury effect on the burn + inhalation situation. Unfortunately, the inhalation only component is absent in almost all the figures afterwards. Therefore, it is uncertain in the increase in immune response, or any other characteristic of burns is a result of the burn or inhalation injury, or both.
4. Figure 4: Inhalation injury gives a higher cytokine release compared all other categories. Any reason for this? How can burns reduce the cytokine release from inhalation injury?
Rapamycin treatment was done to the mice for 7 days. Was it shown previously that this knocked down the mTOR pathway sufficiently? Was it optimized? How did they show molecularly that the mTOR pathway is actually knocked down after 7 days of Rapamycin treatment?
Reviewer 2 Report
Review:
Hall, HR et al. “Characterization of the basal and mTOR-dependent acute pulmonary and systemic immune response in a murine model of combined burn and inhalation injury”
The authors compare their established mouse model for burn injuries with an addition of inhalation injury. It is plausible that a combination of injuries is more severe or more life-threatening but probably fits better to certain human situations. The authors found differences in systemic (plasma) and local (lung) immune mediators and putative pathway activations and tested their models for susceptibility of the mTOR pathway.
The subject is interesting even if no animal model will accurately mirror the human clinical situation or the whole range of injury-related cellular changes but it is of course fair to improve or vary an accepted and established model to display a possible situation. The group is experienced with their model and as an outsider of that explicit field, I have no worries those experiments and pain managements run on high and fair standards.
In general, the manuscript is very long. Even when 10 figures are displayed and no word limitation is set, the text should be more compact and possible proofs should be corrected for little oversights (e.g. name of last author, Plaur1 – Pluar1, …), unclear wordings (”ineffective immune dysfunction” ?), colloquialisms (“deeper dive”), and fillers. I do not believe that 117 references are necessary.
I have a few concerns or questions:
- Is there any explanation why “inhalation only” shows less total lung cells (Fig.1C)? It looks like that “burn” is the driver for immune cell infiltration into the lung. There is even no significant difference between “burn” and “B+I”.
- The text indicates for Fig.1D a “day 14” post injury analysis but there is none.
- What does “data are from 3 independent experiments” in the Fig.1 legend really mean, when each column represents n=6 mice? Was the whole setting displayed in Fig.1B-D repeated three times?
- It seems there is a massive increase of macrophages in the BALF 14 days after simply intubation (sham). When neutrophils show a stable count, macrophages increase from 50.000 cells to more than 2.000.000. Is that effect known for the model used?
- When a protein is analyzed, a representative picture of an iNOS-positive population from FACS-analysis (sham vs. B+I) could be helpful to better catch the analysis of the Median Fluorescence Intensity for iNOS expression. Alternatively, an additional Western-Blot comparing the different treatments could indicate the iNOS expression in harvested macrophages or neutrophils. Again, does each column represent the analysis of n=6 mice? Is it one experiment out of 3 (so, in the statistics would be n=18 for each column)? This is probably not right but unclear to me.
- The data for mTOR are confusing or inconsistent. If IL-10 is decreased in “B” (Fig.8B left) upon systemic Rapamycin application, why it is not in “B+I” (Fig.8B right)? The concluded hyper-proinflammatory response, if IL-10 is the marker, holds not true for the “B+I” model. It’s the same, and even vice versa, for the BALF samples. So, mTOR is on one side involved in pro-inflammatory processes (lung) and on the other side in anti-inflammatory processes (systemic). The mTOR pathway is important to reduce bacterial infections in acute phases but in later stages the mTOR pathway is involved in higher bacterial susceptibility. How is that explainable? At least the cell types responsible for this possibility would be great to know (immunohistochemistry, phagocytosis-assays ± rapamycin, …). There is no mTOR activation shown somewhere (e.g. in isolated macrophages etc.) and every conclusion is based on a global rapamycin effect. The idea presented in the discussion is not understandable. It makes no sense to me if an early rapamycin treatment is necessary to get the later benefit if that early treat has such a worse effect on survival and bacterial susceptibility. Maybe, at least to get the point how mTOR might be involved in the different aspects, it could be presented in a scheme.
The manuscript tries to encompass many data. Personally I think the paper is still on a stage to be a draft and need to be shorten. Many passages are overloaded with too many thoughts drifting away from a red line or a clear point. The scientific line can be interesting showing different impacts in the animal models – at different times and different treats. But this makes the things very complex and increases the need to fit the text step by step with the figures. At the moment it needs a major revision to get understandable or even additional experiments underlining the mTOR aspects. This is not clear at all and it seems to be over-interpreted at this stage. Maybe a data set solely describing the “B+I” model could become more clear.
Round 2
Reviewer 1 Report
The comments are addressed by the authors.
Reviewer 2 Report
Ok